# Acute Zonal Occult Outer Retinopathy (AZOOR) Results from a Clinicopathological Mechanism Different from Choriocapillaritis Diseases: A Multimodal Imaging Analysis

**DOI:** 10.3390/diagnostics11071184

**Published:** 2021-06-29

**Authors:** Carl P. Herbort, Ilir Arapi, Ioannis Papasavvas, Alessandro Mantovani, Bruno Jeannin

**Affiliations:** 1Retinal and Inflammatory Eye Diseases, Centre for Ophthalmic Specialised Care (COS), Clinic Montchoisi Teaching Centre, 1006 Lausanne, Switzerland; i.s.papasavvas@gmail.com (I.P.); beabrunocos@gmail.com (B.J.); 2Department of Ophthalmology, University Hospital Centre “Mother Teresa”, 1000 Tirana, Albania; arapi_ilir@hotmail.com; 3Department of Ophthalmology, Valduce Hospital, 22100 Como, Italy; aless.mant@gmail.com

**Keywords:** acute zonal occult outer retinopathy (AZOOR), choriocapillaritis, indocyanine green angiography (ICGA), blue light fundus autofluorescence (BL-FAF), spectral domain optical coherence tomography (SD-OCT), photoreceptors

## Abstract

Background and aim: AZOOR is a rare disease characterized by loss of zones of outer retinal function, first described by J Donald Gass in 1993. Symptoms include acute onset photopsias and subjective visual field losses. The syndrome is characterized by a normal fundus appearance, scotomas and electroretinographic changes pointing towards outer retinal dysfunction. Evolution, response to immunosuppressive treatment and outcome are difficult to predict. The aim of this small case series was to identify the morphological changes and sequence of events in AZOOR thanks to multimodal imaging. Methods: Charts of AZOOR patients seen in the Centre for Ophthalmic Specialized care (COS, Lausanne, Switzerland) were analyzed by multimodal imaging including fundus photography, fluorescein angiography (FA), indocyanine green angiography (ICGA), blue light fundus autofluorescence (BL-FAF) and spectral domain optical coherence tomography (SD-OCT) in addition to a complete ophthalmological examination including visual field testing and microperimetry, as well as OCT angiography (OCT-A) and ganglion-cell complex analysis when available. Cases and Results: Three AZOOR patients with a mean follow-up of 47 ± 25.5 months were included following the clinical definitions laid down by J Donald Gass. The primary damage was identified at the level of the photoreceptor outer segments with an intact choriocapillaris and retinal pigment epithelium (RPE) layer, these structures being only secondarily involved with progression of the disease. Conclusion: Although AZOOR has often been included within white dot syndromes, some of which are now known to be choriocapillaris diseases (choriocapillaritis entities), our findings clearly commend to differentiate AZOOR from entities such as MEWDS (Multiple evanescent white dot syndrome), APMPPE (Acute Posterior Multifocal Placoid Pigment Epitheliopathy), MFC (Multifocal Choroiditis) and others, as the damage to photoreceptors is primary in AZOOR (a retinopathy) and secondary in choriocapillaritis (a choriocapillaropathy).

## 1. Introduction

Acute zonal occult outer retinopathy (AZOOR) was first described by J.D. Gass in the Donders lecture in 1992 and published in 1993 [1] and a more extensive article was published in 2002 including 51 cases [2]. He described a disease characterized by the sudden onset of subjective scotomas and photopsias due to loss of areas of outer retina with a normal fundus aspect. Agarwal, the co-author of the 2002 article, precisely characterized the disease in the book chapter she wrote in the textbook Uveitis, Text and Imaging [3]. AZOOR affects young to middle aged patients, is largely predominant in women, and starts with an acute onset of visual field defect in one or both eyes associated with photopsias, decrease of contrast sensitivity and photophobia. The fundus is essentially normal at presentation, but later can present pigment clumping with progression of retino-choroidal atrophy. Visual field defects remain at best stable when the disease stabilizes within 4–6 months, but can also progress in a proportion of patients. The etiology is speculative including the hypothesis of the involvement of an unknown infective viral agent with subsequent autoimmune alteration of the photoreceptors. Treatment is empiric, including systemic steroids and/or immunosuppressive agents [3]. Gass speculated that AZOOR was part of a spectrum of diseases including multiple evanescent white dot syndrome (MEWDS), acute idiopathic blind spot enlargement (AIBSE), multifocal choroiditis (MFC) and other sub-entities that used to be grouped under the term of “White Dot Syndromes” and are now known to be diseases of the choriocapillaris [1,2,4]. This assumption by Gass was put forward as the patient characteristics of these inflammatory choriocapillaropathies were similar to those of AZOOR patients and because there was a number of reports in which choriocapillaritis patients were subsequently found to present AZOOR [2]. Reports on such associations continue to be published indicating that patients can be exposed to both types of diseases which does not necessarily imply that they are caused by the same mechanisms but probably represent different disease types caused by different physiopathologies [5,6,7]. The goal of the present report was to perform a multimodal imaging analysis of three cases of AZOOR (Table 1) and so determine its clinicopathology. We used the following imaging devices: SPECTRALIS^®^ Heidelberg^®^ OCT (Heidelberg Engineering GmbH, Heidelberg, Germany), AngioVue^®^ Optovue^®^ OCT-A (Optovue, Fremont, CA, USA), Octopus 900 Haag-Streit^®^ Visual Field (Haag-Streit Diagnostics, Bern, Switzerland), TRC-NW400 Topcon^®^ retinal camera (Topcon, Capelle aan den IJssel, The Netherlands).

## 2. Case Reports

### 2.1. Case 1

A 36-year-old woman was sent for a second opinion. In her past ocular history, she had been treated with systemic corticosteroids for an alleged optic neuritis OD (Oculus dexter). She complained of blurry vision and an annular scotoma OD. The electroretinogram had shown diffuse loss of rod photoreceptors OD and was normal OS (Oculus Sinister).

At presentation her visual acuity was 0.8 OD and 1.0 OS (Snellen chart). There was no inflammation on either side and laser flare photometry values were within normal limits showing no subclinical inflammation. Fundus examination on both sides was essentially normal.

Octopus^®^ visual field testing (Haag-Streit Diagnostics, Bern, Switzerland) showed a ring scotoma OD, and normal OS. Microperimetry was decreased OD (410/560) and within normal limits OS (480/560).

FA showed discreet late ring hyperfluorescence around the fovea OD and at the inferior border of the disc OS and along the superior temporal arcade OS. (Figure 1) ICGA showed no hypofluorescent areas ODS indicating normal and perfused choriocapillaris (Figure 1).

Blue light fundus autofluorescence (FAF) showed hyperautofluorescence of the whole posterior pole OD, except for a central macular area that was normal OD (Figure 2). On the left side there was a peripapillary area of hyperautofluorescence demarcated by a distinctly hyperfluorescent line (Figure 2).

In both eyes the areas of hyperautofluorescence corresponded to the loss of the photoreceptor outer segment line on SD-OCT. Taken together, (1) when compared with FA and ICGA, the areas of hyperautofluorescence corresponded in both eyes to discreet late FA hyperfluorescence and to retained ICGA fluorescence, indicating that there was choriocapillaris integrity and perfusion; (2) combined FAF, FA and ICGA hyperfluorescence indicated a decreased masking effect because of altered or absent screen due to damage and/or loss of photopigment following damage/loss of the outer segments of photoreceptors. The patient planned a pregnancy. Therefore, no immunosuppressive therapy was initiated. After six years of follow-up, the situation in the left eye had remained stable. On the right-side visual acuity had decreased to 0.6, the ring scotoma had slightly increased (Figure 3) but the retinal deterioration was particularly obvious on microperimetry, decreasing from a score of 410/560 to 342/560 with points on the outer part of the central preserved zone showing the sharpest drop of score. (Figure 4) Visual field and microperimetry remained normal in the left eye.

Multimodal imaging findings remained stable in the left eye. On the right side, the peripheral fundus took the aspect of a pseudo retinitis pigmentosa with pigment clumps (Figure 5).

FA late hyperfluorescence in the periphery had increased substantially over the whole fundus due to atrophic changes of the RPE/choriocapillaris complex which corresponded to hypofluorescent areas on ICGA. (Figure 5) The pattern of FAF OD showed an increase of hypoautofluorescent spots in the periphery and within the hyperautofluorescent ring around the preserved central area indicating atrophic changes of the RPE/choriocapillaris complex (Figure 6, white arrows) and corresponding to hypofluorescent spots on ICGA. (Figure 5) Immediately around the preserved central area the more hyperautofluorescent rim was interpreted to be due not only to the loss of photoreceptor pigment screen but also to the accumulation of lipofuscin material in RPE cells in an evolving disease with centripetal extension of the process. (Figure 6) The FAF aspect in the left eye had not changed notably. This case fulfilled the characteristics of AZOOR. The multimodal imaging findings clearly indicated that the pathology resided at the level of the retina with a preserved choriocapillaris followed only later by areas of secondary choriocapillaris loss.

### 2.2. Case 2

This 56-year-old woman was sent to our center for a scotoma in her right eye that had first been noted 17 years prior to her visit. Nine years earlier the scotoma had increased in OD which led her doctors to discontinue the hydroxy-chloroquine therapy she had been given during 17 years for the treatment of systemic lupus erythematosus. Two years prior to her visit, her doctors had started an immunosuppressive treatment for her systemic disease with mycophenolic acid, 720 mg BID (Myfortic^®^, Novartis, Cambridge, MA, USA). One year prior to her visit a scotoma had also appeared in her left eye. Electrophysiology had shown diffuse loss of rod photoreceptors more pronounced in OD than OS.

At presentation her best corrected visual acuities were 1.0 OD and 0.8 OS (amblyopic eye) (Snellen chart). Intraocular pressure was 14 mmHg ODS. There was no anterior inflammation on either side visible at the slit lamp, but laser flare photometry (LFP) showed slight subclinical inflammation with values of 5.8 photons/milliseconds (ph/ms) OD and 8.9 ph/ms OS. Fundus examination showed a pale ring around the fovea OD (Figure 7A) and was normal OS (Figure 7B top right triad).

Octopus^®^ visual field testing (Haag-Streit Diagnostics, Bern, Switzerland) showed a scotoma temporal to the fovea OD and normal OS. Microperimetry was within normal limits OD (448/560) and OS (504/560). Ganglion cell layer measurement showed an almost complete loss outside of the fovea temporally. FA showed faint hyperfluorescence corresponding to the C-shaped hyperautofluorescent area on FAF OD (see below). (Figure 7, FA 1) Within this area along the temporal superior arcade there was a zone of brighter hyperfluorescence (window defect) indicating chorioretinal atrophy. (Figure 7, FA 1) FA OS was normal (not shown). In the periphery of the right eye, FA showed supero-temporal vasculitis (Figure 7G, yellow arrow). Late ICGA frames showed hyperfluorescence corresponding to the C-shaped hyperautofluorescent areas on FAF, except in the area along the superior temporal arcade which was hypofluorescent corresponding to hypoautofluorescence on FAF and hyperfluorescence on FA (Figure 7D, crimson arrow), all images indicating chorioretinal atrophy in these areas (Figure 7C–F, crimson arrow).

Fundus autofluorescence (FAF) showed an extensive C-shaped hyperautofluorescent area around the fovea with dots of hypoautofluorescence within this zone along the superior temporal arcade OD. (Figure 8A) On the left side there were faint areas of hyperautofluorescence parapapillary inferiorly and supero-temporally to the fovea. (Figure 8A FAF OS 1) In both eyes the FAF hyperautofluorescence corresponded to the loss of photoreceptors on SD OCT, extensive in OD (Figure 8C) and discreet OS (not shown).

The areas of hyperautofluorescence on FAF, slightly hyperfluorescent on FA and hyperfluorescent on ICGA, indicated that the choriocapillaris was unharmed (except along the superior temporal arcade). SD-OCT showed loss of the photoreceptor outer segments in these areas, explaining the FAF autofluorescence due to the loss of the masking effect of the RPE lipofuscin by the loss of photoreceptor layer. The analysis of the ganglion cell complex by Optovue OCT showed that there was a substantial ganglion cell loss in the areas of photoreceptor loss delineated by the different imaging modalities (FAF, FA, ICGA, SD-OCT). (Figure 9) The left eye was within normal limits, as far as functional and morphological parameters were concerned.

The patient was treated with mycophenolic acid (Myfortic^®^, Novartis, Cambridge, MA, USA), 720 mg BID and infliximab (Remicade^®^, Janssen Biotech Inc., Drive Horsham, PA, USA) 5 mg/kg every 6 weeks and after 18 months evolution was stable, as far as FAF, FA, ICGA and SD-OCT was concerned OU. (Figure 7, FA2, ICGA 2 and Figure 8, FAF OD 2, OCT OD 2).

This case illustrates clearly that the pathology in AZOOR is primarily at the level of the photoreceptors and not due to choriocapillaritis, which can be involved secondarily during progression of AZOOR.

### 2.3. Case 3

This 44-year-old woman was sent to us for a second advice by her uveitis specialist. She was consulted for left photopsias, decreased contrast sensitivity, photophobia and scotomas without any overt intraocular inflammation. FA showed bilateral retinal vasculitis. SD-OCT showed loss and/or disruption of the photoreceptor outer segment layer, the ICGA was normal—in particular no choriocapillaris circulation voids were detected. A neuroophthalmological cause had been excluded. Electrophysiological investigation detected photoreceptor dysfunction ODS with reduced amplitudes. FAF showed left hyperautofluorescence in the posterior pole and one year later the right eye was involved in a similar fashion. Adalimumab (Humira^®^, AbbVie, Chicago, IL, USA), 40 mg every 2 weeks was started and 6 months later Mycophenolate Mofetil (Cellcept^®^, Genentech, San Francisco, CA, USA), 2 g per day was added. When the disease progressed despite dual immunosuppressive therapy, a second opinion was solicited. She had gone through numerous episodes of tonsillitis and her ASLO titers (antistreptolysins) had been elevated several times.

At presentation her visual acuity OD was 0.2 sc for far and 0.1 sc for near, and for OS it was 0.4 for far and 0.2 for near. Intraocular pressure was 15 mmHg ODS. There was a subclinical anterior segment inflammation with laser flare meter values of 12.1 ph/ms OD and 12.4 ph/ms OS. There were rare cells in the vitreous ODS.

Fundus examination showed faint discolored rings around the foveas ODS. (Figure 10) FA showed a diffuse peripheral retinal vasculitis on 360° ODS. In the posterior pole there was faintly hyperfluorescent ring ODS. The ICGA of the posterior pole ODS did not show hypofluorescent areas indicating integrity of the choriocapillaris except for two interpapillo-macular dark dots OS (Figure 10).

Blue light FAF showed hyperautofluorescence of the whole posterior poles ODS, with the exception of a preserved central foveolar area. The hyperautofluorescence was due to the loss of photoreceptor outer segments, which was very extensive as shown by SD-OCT with only a limited central area with preserved photoreceptors (Figure 11a). These changes had increased drastically when compared to the situation 4 years earlier (Figure 11b).

Such an extensive loss of photoreceptor translated into severely impaired visual fields ODS (Figure 12). Microperimetry was characterized by a very low score OD of 174/560 and was not reliable for OS (Figure 13).

It was decided to add cyclosporine (4.5 mg/kg/day) as a third immunosuppressive agent but the situation slightly deteriorated further at 4 months follow-up.

## 3. Results-Summary

Multimodal imaging analysis of AZOOR identified the outer retina as the structure primarily involved, as already hinted by Gass who, adequately, spoke of an outer retinopathy [2] (Table 2). There is damage and loss of photoreceptor outer segments in the posterior pole and beyond (with sparing of the central foveal area), clearly shown on SD-OCT. (Figure 4, top right) This process causes the following imaging characteristics: (1) a faint discolored halo around the fovea on fundus photography (Figure 4, top left); (2) a slight halo of late discreet FA hyperfluorescence around the fovea (Figure 4 bottom left); (3) a conserved ICGA fluorescence being hyperfluorescent in the area of photoreceptor loss (Figure 4, bottom middle); (5) FAF hyperautofluorescence in the area of SD-OCT photoreceptor loss (Figure 14E). When the lesions progress, chorioretinal atrophy develops (arrows in C, D, E of Figure 14), hyperfluorescent on FA (window effect), dark on ICGA (choriocapillaris drop-out) and dark on FAF (loss of RPE cells).

OCT angiography did not contribute significant additional information in the two patients in whom it was performed. It simply showed integrity of the choriocapillaris, which was already demonstrated on ICGA (Figure 15).

While imaging findings were consistent in all three patients, the evolutionary patterns, difficult to predict, differed in the 3 patients. In our series it went from minimal progression without treatment in patient 1, to clinical control of disease with dual immunosuppressive therapy in patient 2, and to progression despite triple immunosuppressive therapy in patient 3. Disease was bilateral in all 3 patients with subclinical non-progressing evolution in the fellow eye in two patients.

## 4. Discussion

AZOOR was included by Gass in the spectrum of “white dot syndromes” including MEWDS, AIBSE, Multifocal choroiditis and others now known to be inflammatory diseases of the choriocapillaris. Reports including AZOOR in this spectrum, lumping it together with choriocapillaritis entities, continued to be published up to recently after Gass put forward this hypothesis [4,6,8,9,10,11,12]. Thanks to multimodal imaging it is now possible to clearly distinguish and split AZOOR from choriocapillaritis entities. The latter have their origin in inflammatory non perfusion of the choriocapillaris, which is precisely delineated thanks to indocyanine green angiography (ICGA) [13,14,15].

In contrast, in AZOOR the structure primarily affected is the outer retina as was hypothesized by Gass. With the availability of multiple imaging modalities, the physiopathology of the disease, photoreceptor damage, could be precisely demonstrated [16,17,18]. However, the choriocapillaris can be involved secondarily, as a collateral damage as shown on Figure 4 [19].

Our small case series verified this disease mechanism. Combined analysis using FA, ICGA, BL-FAF and SD-OCT pointed towards the photoreceptors as the site of the primary lesion process while choriocapillaris was preserved and perfused, thus differentiating AZOOR from choriocapillaritis entities. One element that our AZOOR series had in common with choriocapillaritis was a mean myopic refraction of −2.5 diopters.

The two types of conditions have been described to occur sometimes in the same patient [7,20,21]. This probably represents more than a chance occurrence and it is possible that one disease can predispose to the other or vice versa.

Although the characterization of typical AZOOR cases is quite precise, atypical forms involving the outer retina can occur, difficult to classify within the well-defined AZOOR criteria.

Retinal vasculitis was present in two of our patients, unilateral faint peripheral in one eye in patient 2 and diffuse pronounced and bilateral in patient 3, a finding which has been described previously [17].

Bone spicule pseudo retinitis pigmentosa evolution was noted in one patient with bone spicule lesions after 6 years of evolution in patient 1 which has also been described previously [17]. The most rewarding imaging modalities for AZOOR were without doubt SD-OCT showing photoreceptor outer segment damage and, even more so, blue light FAF delineating the extension of lesions [22,23,24].

As AZOOR is a rare disease, no therapeutic trials have been conducted and treatment is empirical. Spontaneous remission has been described [25]. Favorable evolution without treatment in 3/5th of eyes and after systemic corticosteroids in 2/5th of eyes has been reported in a large Japanese study including 52 eyes [26]. Corticosteroids have been reported to be effective by systemic administration [27], pulse intravenous corticosteroid therapy [28] or intravitreal administration [29,30]. Nonsteroidal immunosuppression has been proposed [17,31]. Biologic agents have resulted in mixed results, AZOOR having responded to the anti-TNF-agent adalimumab [5] but not to abatacept [32]. Evolution and response to treatment is difficult to predict. The presence or absence of anti-retinal antibodies does not seem to influence treatment and outcome [33]. However, thanks to multimodal imaging close follow-up is possible.

## Figures and Tables

**Figure 1 diagnostics-11-01184-f001:**
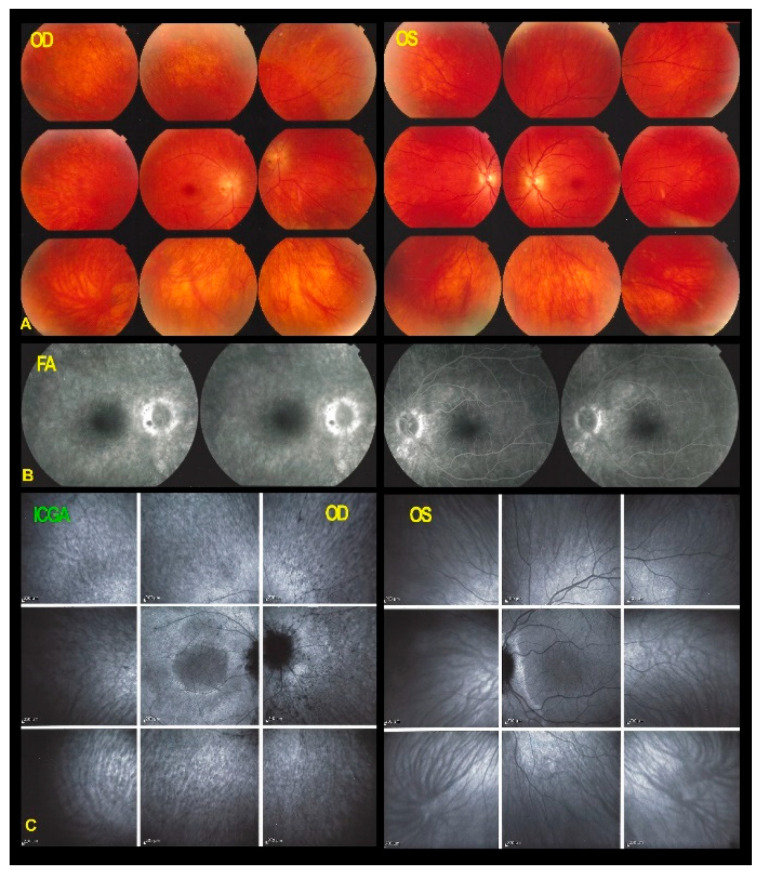
Fundus photographs, FA & ICGA frames ODS in AZOOR patient 1 at presentation. (**A**) No manifest changes are visible on the fundus photographs ODS. (**B**) FA shows discreet late hyperfluorescence in the posterior pole around the fovea OD and around the inferior part of the optic disc OS and along the superior temporal vascular arcade OS. (**C**) ICGA ODS (late angiographic frames) does not show capillary drop-out.

**Figure 2 diagnostics-11-01184-f002:**
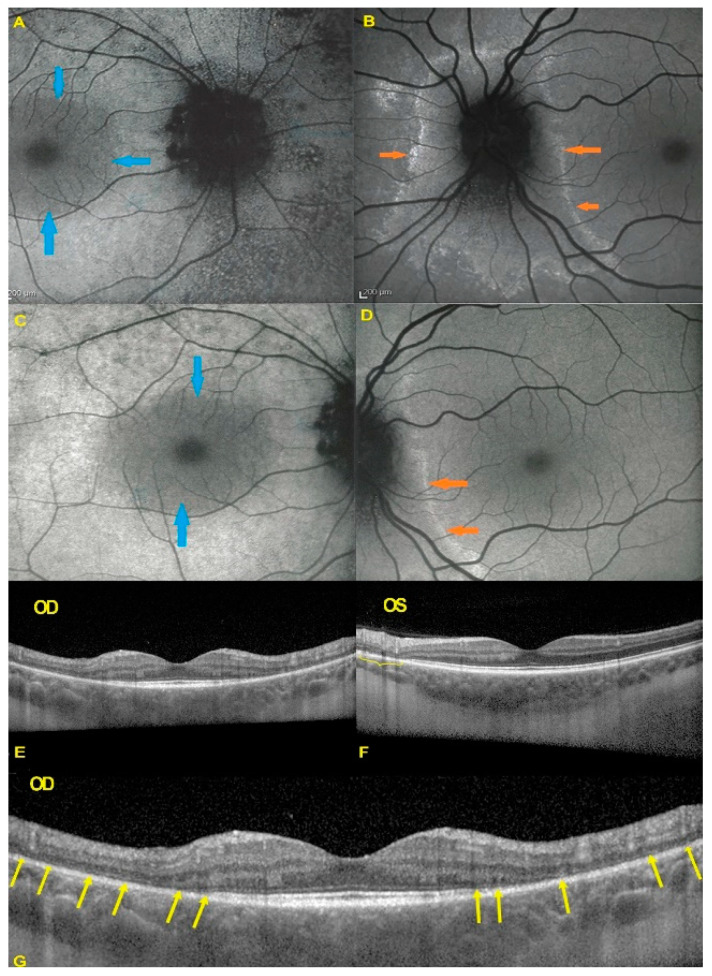
BL-FAF & SD-OCT in AZOOR patient 1 at presentation. (**A**) Increased FAF of the whole posterior pole OD (**A**,**C**) except the central foveolar and peri-foveolar area (blue arrows) explained by the loss of the screen of the photoreceptor outer segments as shown on the SD-OCT (**G**) sections (yellow arrows). On the left eye in FAF (**B**,**D**) there is peripapillary hyperautofluorescent demarcating line (orange arrows) corresponding also to a limited area of loss of photoreceptor outer segments (**E**–**G**, yellow bracket in **F**, yellow arrows in **G**).

**Figure 3 diagnostics-11-01184-f003:**
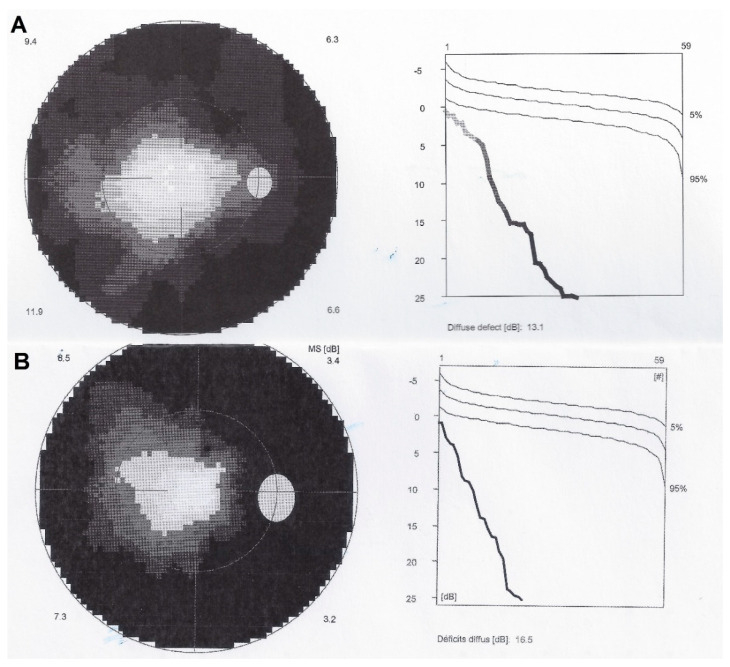
Visual fields OD in AZOOR patient 1. Shows tubular visual field at presentation (**A**) and increase of annular scotoma after 6 years of follow-up (**B**).

**Figure 4 diagnostics-11-01184-f004:**
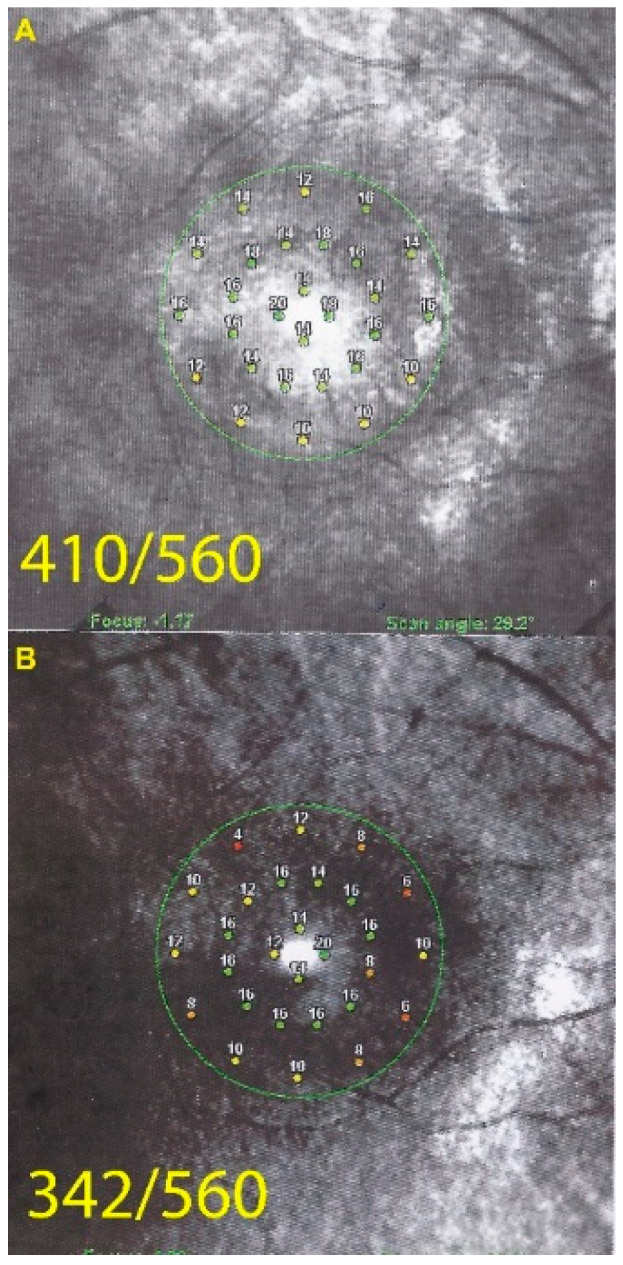
Microperimetry OD in patient 1 with AZOOR at presentation (**A**) and after 6 years of follow-up (**B**). Decrease of microperimetry score especially among the peripheral measurement points after 6 years of evolution indicating slow progression of the process.

**Figure 5 diagnostics-11-01184-f005:**
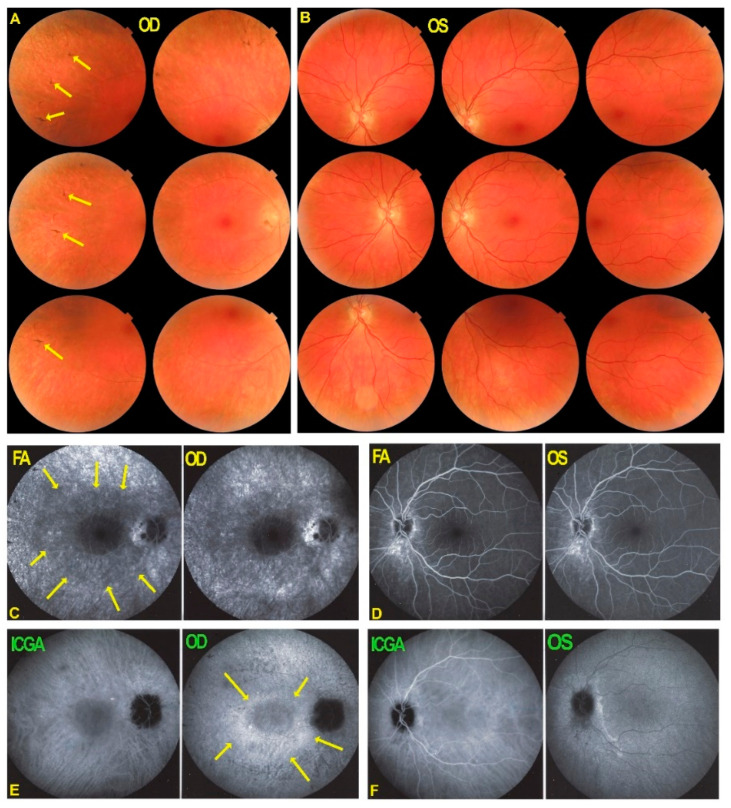
Fundus photographs, FA and ICGA frames ODS in AZOOR patient 1 after 6 years of evolution. The right fundus (**A**) shows an evolution towards pseudo retinitis pigmentosa with a faint bone spicule aspect (yellow arrows). No other manifest changes are visible on the fundus photographs (**A**,**B**). FA shows discreet late annular hyperfluorescence around the fovea OD (**C**, yellow arrows) and increased late hyperfluorescence (window effect/staining) in the more peripheral areas due to chorioretinal atrophy. FA in OS shows hyperfluorescence inferior to the optic nerve (**D**). (**E**,**F**) ICGA shows maintained late fluorescence around the normal central area (**E**, yellow arrows), indicating preserved choriocapillaris.

**Figure 6 diagnostics-11-01184-f006:**
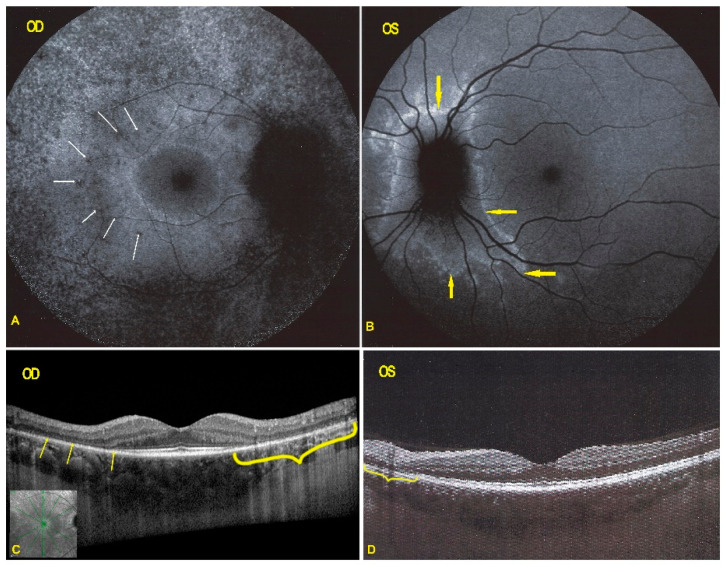
Follow-up of BL-FAF and SD-OCT in AZOOR patient 1 after 6 years of evolution. (**A**) FAF OD shows a rim of more pronounced hyperautofluorescence around the preserved central area indicating centripetal progression of the process as well as apparition of dark areas in the posterior pole hyperautofluorescence rim (white arrows) indicating secondary atrophic evolution. FAF OS (**B**) shows a demarcation hyperautofluorescent line around the optic nerve (yellow arrows). SD-OCT (**C**,**D**) shows the loss of photoreceptor outer segments (yellow brackets, yellow arrows).

**Figure 7 diagnostics-11-01184-f007:**
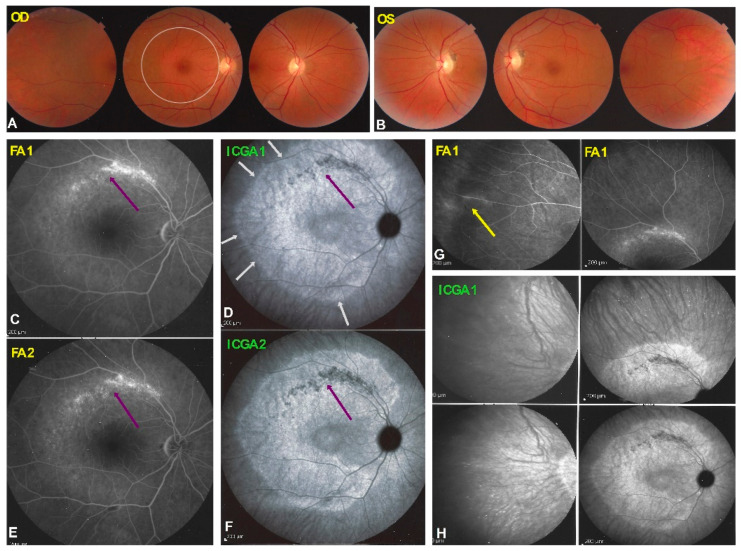
Fundus photographs, FA and ICGA in patient 2 with AZOOR at presentation and follow-up. (**A**) Fundus OD shows pale ring around fovea (white circle) while fundus OG (**B**) is normal. FA OD shows a discreetly hyperfluorescent ring around fovea with an arcuate form of bright hyperfluorescence along superior temporal arcade OD indicating chorioretinal atrophy (**C**, crimson arrow) which slightly increased after 18 months (**E**, crimson arrow). (**D**) ICGA in OD showed a reinform, C-shaped hyperfluorescence around the fovea indicating preserved choriocapillaris (white arrows), except in the area of atrophy (hyperfluorescent on FA) (**D**, crimson arrow). The latter area slightly increased after 18 months (**F**, crimson arrow). (**G**) FA showed discreet peripheral vasculitis (yellow arrow). (**H**) The extension of ICGA hyperfluorescence is shown.

**Figure 8 diagnostics-11-01184-f008:**
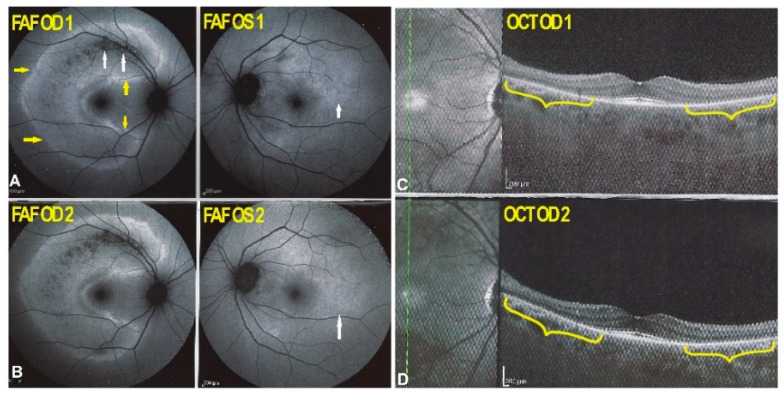
BL-FAF & SD-OCT in AZOOR patient 2 at presentation and follow-up. Blue light FAF ODS at presentation (**A**)and after 18 months (**B**) showing OD a reniform (C-shaped, yellow arrows) hyperfluorescence due to loss of outer segments of photoreceptors except in the arcuate area along the superior temporal arcade (white arrows) corresponding to FA hyperfluorescence and ICGA hypofluorescence due to chorioretinal atrophy. The bl-FAF of the left eye shows discreet hyperautofluorescence temporal superior to the fovea (**A**,**B**, FAF OS1 and FAF OS2 white arrow). SD-OCT OD shows ± identical loss of the photoreceptor outer segment line at presentation (**C**) and after 18 months (**D**).

**Figure 9 diagnostics-11-01184-f009:**
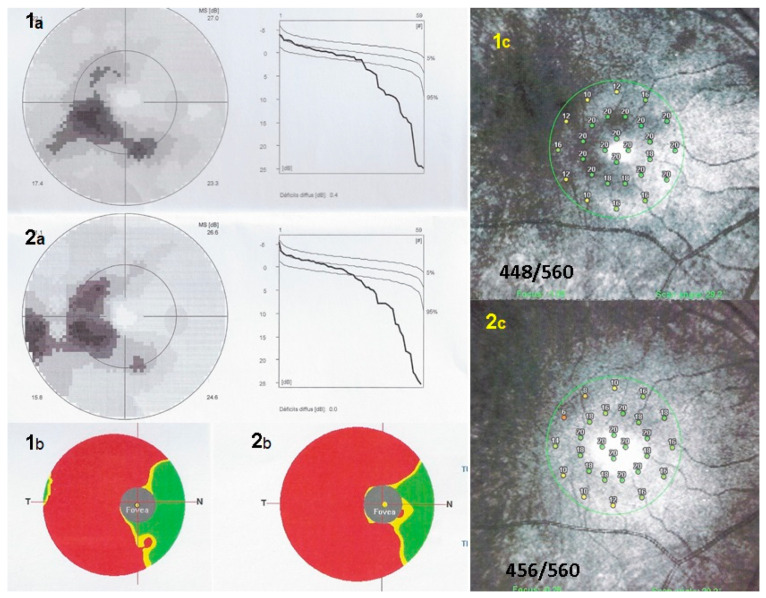
Visual field findings, ganglion cell layer and microperimetry in case of AZOOR at presentation and after 18 months of follow-up in patient 2. Visual field (**1a**) showed a limited scotoma also seen after 18 months (**2a**). Ganglion cell layer (**1b**,**2b**) shows a substantial loss outside the fovea (corresponding to scotoma in **1a**) which increased slightly after 18 months (**2b**). Microperimetry shows a score of 448 OD at presentation (**1c**) which did not decrease after 18 months of follow-up (**2c**).

**Figure 10 diagnostics-11-01184-f010:**
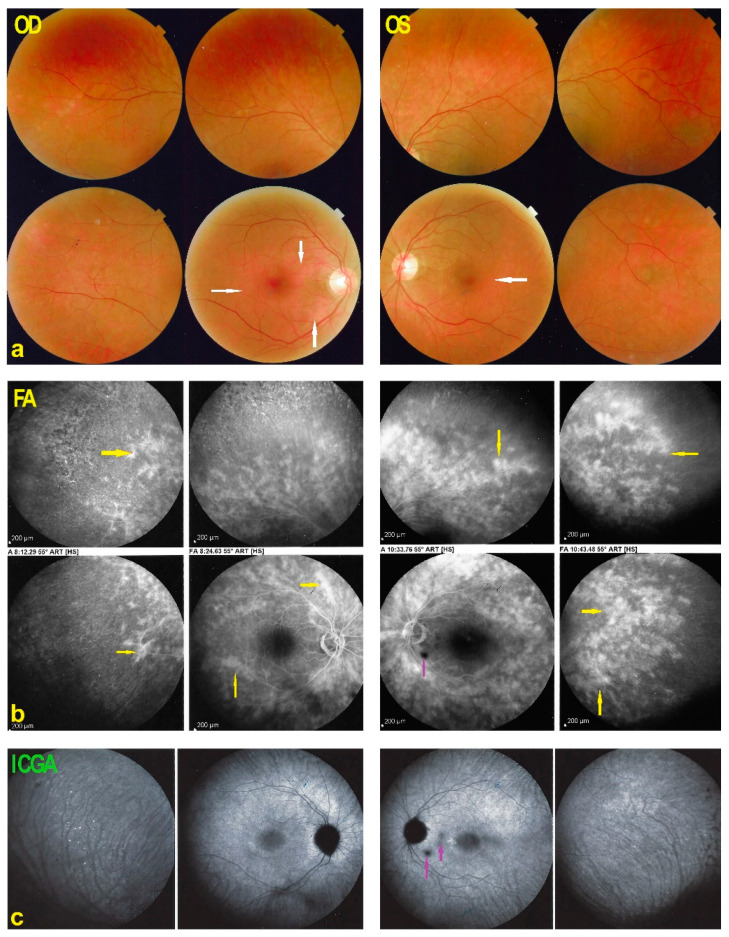
Fundus, FA and ICGA of AZOOR patient 3 at presentation. (**a**) Fundus shows bilateral discolored rings around fovea (white arrows). (**b**) FA shows extensive peripheral retinal vasculitis ODS (yellow arrows). Late ICGA frames present ICGA fluorescence indicating conserved choriocapillaris except for two interpapillo-macular dark dots (**c**, crimson arrow) in the left eye also present on the FA image (**b**, crimson arrow).

**Figure 11 diagnostics-11-01184-f011:**
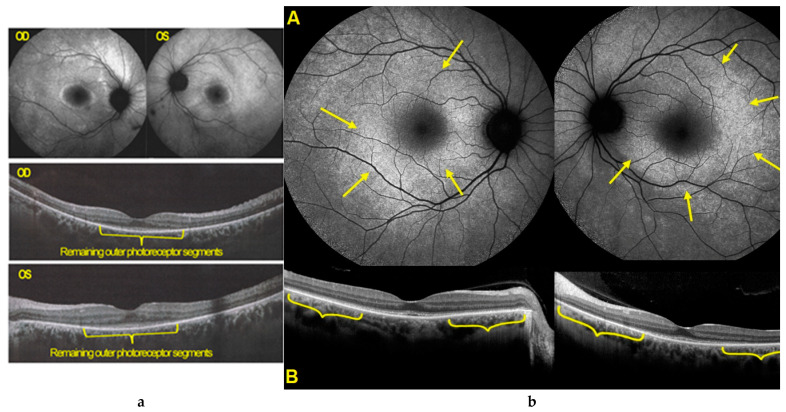
(**a**). BL-FAF and SD-OCT of AZOOR patient 3 at presentation. Diffuse FAF hyperautofluorescence of the posterior poles ODS due to loss of screen of photopigment present in the photoreceptor outer segments lost in the whole posterior pole except in the central foveolar area surrounded by a more hyperautofluorescent ring probably indicating centripetal progression of the process. Only a very central area of conserved outer segments can be seen ODS (yellow brackets). (**b**). BL- FAF and SD-OCT of AZOOR patient 3, 4 years earlier. Well visible annular hyperautofluorescence (**A**, yellow arrows) with partial to complete loss of outer segment line (**B**, yellow brackets).

**Figure 12 diagnostics-11-01184-f012:**
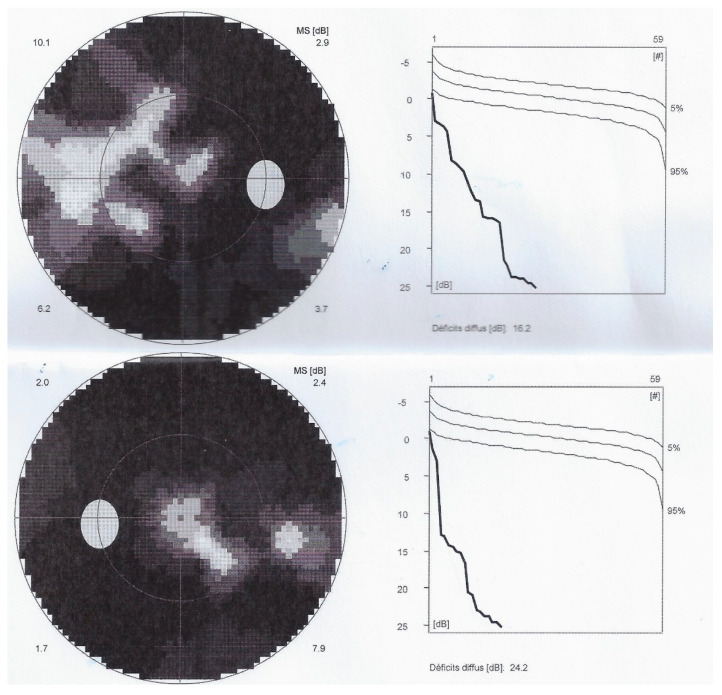
Visual fields ODS in AZOOR patient 3 at presentation. Severe visual fields loss ODS.

**Figure 13 diagnostics-11-01184-f013:**
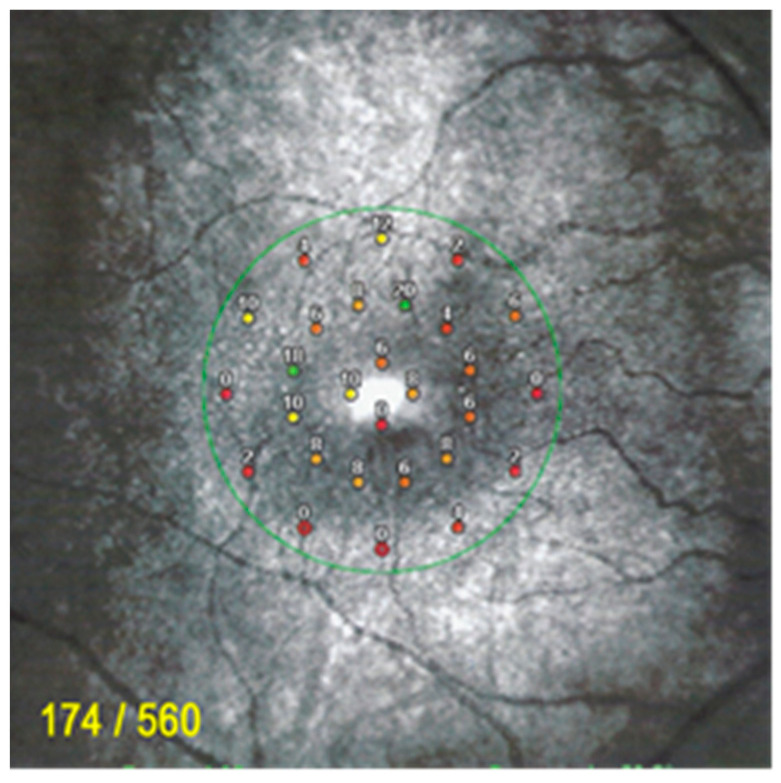
Microperimetry OD in AZOOR patient 3 at presentation. Severe loss of score reduced to 174/560.

**Figure 14 diagnostics-11-01184-f014:**
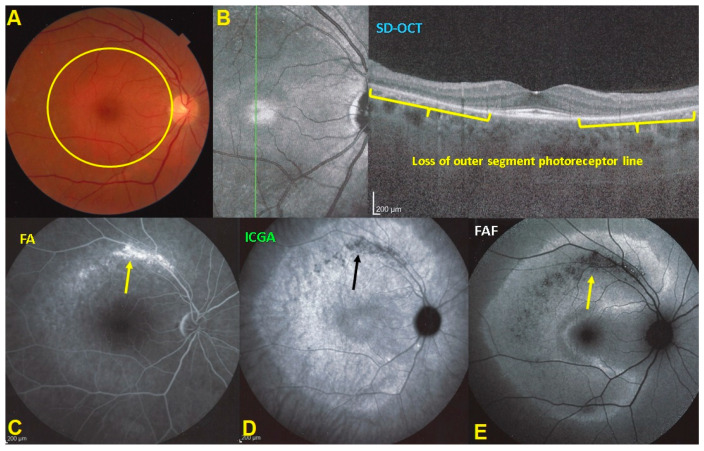
Imaging illustration of the clinicopathology of AZOOR. (**A**) Fundus shows a pale discolored halo around the fovea that retains a normal color (yellow circle) due to loss of photoreceptor photopigment. (**C**) FA OD shows the same halo of discreet hyperfluorescence due to photopigment loss and an area of bright hyperfluorescence (window effect) along superior temporal arcade due to chorioretinal atrophy (dark on ICGA—**D**, black arrow- and FAF—**E**, yellow arrow). ICGA (**D**) shows preserved choriocapillaris (except in the arciform area of chorioretinal atrophy-black arrow) with increased fluorescence in the area of loss of the screen of photopigments, which also explains fundus hyperautofluorescence (**E**). SD-OCT (**B**) shows the loss photoreceptor outer segments.

**Figure 15 diagnostics-11-01184-f015:**
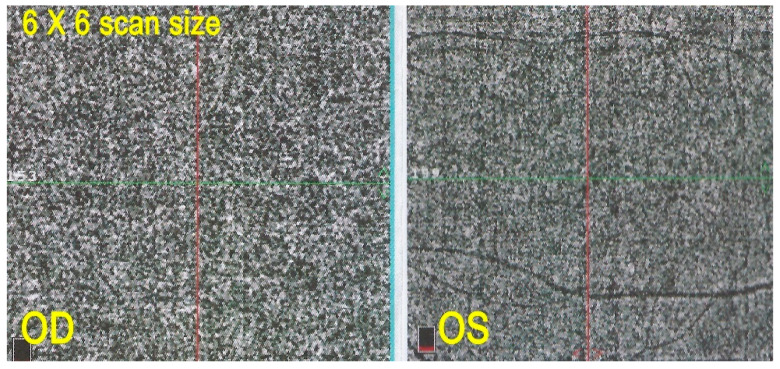
OCT-A in AZOOR patient 2. Capillary plane: no capillary drop out ODS.

**Table 1 diagnostics-11-01184-t001:** Demographics, visual acuity, history and symptoms.

Patients	Age	Gender	Refract. OD	Refract. OS	VA OD	VA OD (F-up)	VA OS	VA OS (F-up)	Past History	Symptoms
Patient 1	36	f	0.5	0.25	0.8	0.6	1	1	Alleged optic neuritis	Blurred Vision
										Subj. annular scotoma
										Migraines
Patient 2	56	f	−0.25	−0.25	1	1	0.8	0.8	Autoimmune Lupus like Disease	Subj. Scotomas OD > OS
										Photophobia
Patient 3	44	f	−2	−2.75	0.2	0.1	0.35	0.25	ASLOS increased	Photopsias
										Dim. Contrast sensitivity
										Photophobia

VA = vision acuity; Refract = refraction; F-up = follow up.

**Table 2 diagnostics-11-01184-t002:** Multimodal imaging of AZOOR and clinicopathological correlations.

BL-FAF	Hyperautofluorescence [increased exposure of RPE lipofuscin following loss of photoreceptor OS]
	Hypoautofluorescence in severely affected areas [chorioretinal atrophy]
ICGA	Preserved fluorescence [indicates integrity of choriocapillaris]
SD-OCT	Loss of photoreceptors outer segment in diseased areas
FA	Faint late hyperfluorescence in diseased areas [indicates loss of photopigment of photoreceptors OS]
	Hyperfluorescent areas early and late [indicates chorioretinal atrophy-window effect)–
Follow-Up	Areas of chorioretinal atrophy; secondary choriocapillaris involvement

BL-FAF = blue light fundus autofluorescence; ICGA = Indocyanine green angiography; SD-OCT = spectral domain optical coherence tomography; FA = fluorescein angiography; OS = outer segment.

## Data Availability

Please refer to corresponding author.

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
