# Peer review of "Acute Zonal Occult Outer Retinopathy (AZOOR) Results from a Clinicopathological Mechanism Different from Choriocapillaritis Diseases: A Multimodal Imaging Analysis"

_diagnostics, 2021, doi:10.3390/diagnostics11071184_

Round 1
Reviewer 1 Report
The authors describe cases for Acute zonal occult outer retinopathy (AZOOR) and use several imaging
methods for analysis. Aim is to study findings to separate the AZOOR from other choriocapillaris diseases.
The cases and multimodal imaging findins are explained in detail and identifies the outer retina as the structure
primarily involved which was the original discovery of Gass in 1992. An overview of multimodal imaging
results in case studies considering clinicopahological correlations
is provided
Introduction lacks the description of the goals in this article.
Details of the imaging devices might be of interest to readers
Some general abbreviations used could be explained when
first time mentioned. like OS, OD
Some english checking needed, some prepositions are missing also some typos in text,
below some examples
presentation ->examination?
line 67 OD and was normal OS
line 84 a ring scotoma OD and was normal OS
line 103 slows
line 105 planed
line 124 Multimodal imaging remained stable in the left eye ->Multimodal imaging findings? remained stable in the left eye
line 134 to accumulation lipofuscin
line 135 in the left eye had nor changed notably
line 147 had increased OD
line 152 loss of rod photoreceptors OD > OS.
line 155 were 14 mmHg OU
line 223 She had consulted for
line 235 OS it was
line 238 There were rare cells in the vitreous cells ODS
line 264 could not be performed reliably OS
Author Response
Reviewer 1
Introduction lacks the description of the goal in this article:
Following sentence was added: “The goal of the present report was to perform a multimodal imaging analysis of three cases of AZOOR and so determine its clinicopathology.”
Details of the imaging devices might be of interest.
We added details on the equipment used
Some general abbreviations could be explained…
Oculus dexter (OD); Oculus sinister (OS)
Some English checking was performed
The detailed corrections indicated were performed (indicated in yellow)
Reviewer 2 Report
The manuscript titled: "Acute zonal occult outer retinopathy (AZOOR) results from a clinicopathological mechanism different from choriocapillaritis diseases: a multimodal imaging analysis" by Carl P. Herbort Jr et al. is a valuable contribution to the field; however the Authors need to address the following points:
1) The Authors in the introduction, lines 48-49, mention that a viral agent could be involved in the pathogenesis of AZOOR. The Authors should suggest what viruses could be involved.
2) Figures and tables are presented in the paper before their descriptions in the text; please insert each figure after you have referred to it in the text. Table 1 (page 13) is not described in the text; Fig 5 (line 290) is missing and the supplementary figure is never reported in the text.
2) Figures are not always clear, including legends. The Authors should clearly indicated by arrows all the abnormalities they are observing and when they refer to the right versus the left eye.
4) Abstract: line 30: it should be MFC, and not MEWDS as it is already mentioned; and the abbreviation should be used in line 311
5) Line 42 : It should be [3]. and not .[3]
Same issue for line 87 and 88 with Figure 1.1 and others in the manuscript. The Authors should make sure to be consistent along all the manuscript. Please remove "(Figure 4)" line 322.
Author Response
Reviewer 2
- The virus is unknown. It is a hypothesis. We add the word unknown in the text.
- Thank you for your remarks. Text is corrected, table 1 reference is added to the text and figure 5 was added. Supplementary figure is figure 1.
- Arrows were added to photos. Eye is indicated in the legends
- MFC is added
- We corrected in the text
Reviewer 3 Report
The manuscript “Acute Zonal Occult Outer Retinopathy (AZOOR) Results from A Clinicopathological Mechanism Different from Choriocapillaritis Diseases: A Multimodal Imaging Analysis” by Herbort et al. is devoted to the addressing of mechanisms of the Acute Zonal Occult Outer Retinopathy (AZOOR) disease. The authors examined 3 cases of this rare disease using multiple optical modalities and identified the utility of these modalities for AZOOR diagnostics (ICGA as the most helpful). In addition to that, they made some observations and conclusions regarding the nature of this disease (a retinopathy rather than choriocapillaritis nature). The article is well-written in general and can be of interest to broad clinical and scientific audiences.
It can be published if the following shortcomings were addressed:
- Line 29-30: Use of undefined abbreviations in the abstract. Also make sure that you don’t double entry MEWDS in “MEWDS, APMPPE, MEWDS”
- Line 65,68: Abbreviations (even obvious ones as OD, OS) should be defined on their first appearance
- Panels of every Figure should be numbered (e.g., a,b, c…) and referred to in captions instead of top-right, etc.
- Figures 1.3 and 3.3 are of unacceptable quality
- Do figures refer to the left or right eye? (for example, Fig 1.3 and 1.4). It should be mentioned in all image captions
- Tables 1 and 2 are of unacceptable quality
Author Response
Reviewer 3
- Definitions of abbreviations were added in the abstract; double entry was deleted.
- Abbreviations such as OD and OS were explained
- Panels were numbered
The side of the eye involved was added in the captions
- Figures 1.3 and 3.3 were improved
- Tables were improved